# Leveraging Early-Stage Robustness in Diffusion Models for Efficient and High-Quality Image Synthesis

**Yulhwa Kim[1], Dongwon Jo[1], Hyesung Jeon[1], Taesu Kim[2], Daehyun Ahn[2], Hyungjun Kim[2], Jae-Joon Kim[1]**
[1]Seoul National University, [2]SqueezeBits Inc.
{yulhwakim, dongwonjo, hjeon2k, kimjaejoon}@snu.ac.kr,
{taesu.kim, daehyun.ahn, hyungjun.kim}@squeezebits.com

## Abstract

While diffusion models have demonstrated exceptional image generation capabilities, the iterative noise estimation process required for these models is compute-intensive and their practical implementation is limited by slow sampling speeds. In this paper, we propose a novel approach to speed up the noise estimation network by leveraging the robustness of early-stage diffusion models. Our findings indicate that inaccurate computation during the early-stage of the reverse diffusion process has minimal impact on the quality of generated images, as this stage primarily outlines the image while later stages handle the finer details that require more sensitive information. To improve computational efficiency, we combine our findings with post-training quantization (PTQ) and introduce a method that utilizes low-bit activations for the early reverse diffusion process while maintaining high-bit activations for the later stages. Experimental results show that the proposed method can accelerate the early-stage computation without sacrificing the quality of the generated images.

## 1 Introduction

Diffusion models [1, 2, 3, 4, 5, 6, 7, 8] for image synthesis have gained significant attention in recent years due to their exceptional proficiency in generating high quality and diverse images. A key feature of the diffusion models is that they gradually dissipate noise from latent variables across a sequence of diffusion steps [1]. However, despite their remarkable image synthesis capabilities, diffusion models suffer from slow image synthesis process. This is primarily caused by the compute-intensive nature of the denoising network [3, 9], which is an integral part of the diffusion model. The iterative process of applying the denoising network across multiple diffusion steps consumes substantial computational resources, resulting in very long synthesis time [6]. As a result, this limitation poses a challenge for real-time applications or scenarios that require quick image synthesis.

Quantization [10, 11, 12] is a widely used technique for improving the computational efficiency of neural networks. Quantization reduces memory usage by converting high-precision floating-point weight/activation values into low-precision integer values [13]. It also enables more efficient processing by leveraging efficient integer operations instead of computationally expensive floating-point operations [14]. As a result, there is a growing research trend focusing on applying quantization techniques to denoising networks. Two notable studies, PTQ4DM [15] and Q-diffusion [16], explored the applications of LAPQ [17] and BRECQ [18], which are commonly used Post-Training Quantization (PTQ) frameworks, to the diffusion models. Denoising networks differ from conventional neural networks as they are iteratively used during the reverse diffusion process, and their output distributions vary with diffusion step. Considering this characteristic, the aforementioned studies primarily explored methods for constructing a calibration dataset to effectively apply PTQ to diffusion

37th Conference on Neural Information Processing Systems (NeurIPS 2023).

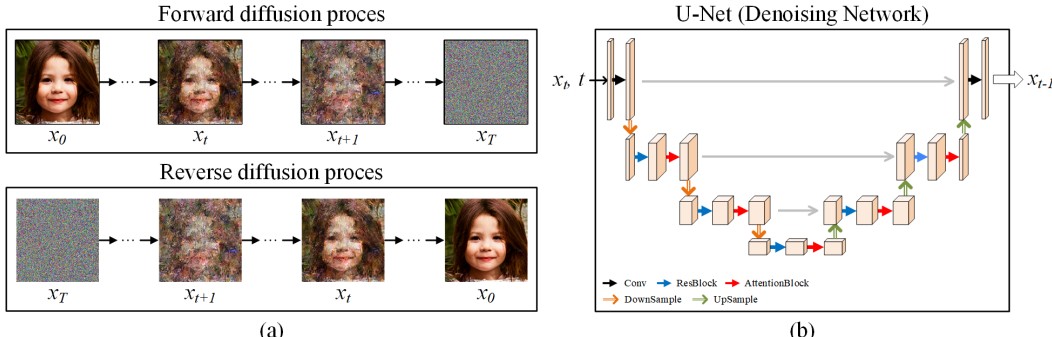

Figure 1: (a) Overview of the diffusion process (b) U-Net architecture of the denoising network

models. While they have successfully achieved a reduction in weight precision to as low as 4 bits, reducing activation bits below 8 bits still remains a challenge.

In this paper, we investigate the computational characteristics of diffusion models during their iterative diffusion steps and present a novel strategy for quantizing diffusion models based on the analysis. Our analysis of the diffusion model reveals the high tolerance of early-stage reverse diffusion in the presence of computational errors. In the reverse diffusion process, the early stage focuses on capturing the outlines of the images, while the later stage refines the details to improve the overall quality of the generated images. As a result, modifications made in the early stages primarily affect the structural aspects of the generated images, while the quality remains mostly unaffected. Based on the observation, we propose leveraging the early-stage robustness observed in the diffusion process to optimize activation bit precision, thereby reducing computational overhead while maintaining the fidelity and diversity of the generated images. By strategically quantizing the activation values in a step-wise manner, we can reduce the effective bit precision of the activations of the denoising networks without sacrificing the overall quality of the generated images.

## 2 Background

### 2.1 Diffusion Models

The generation of an image in diffusion models is represented as a series of diffusion steps (Fig. 1(a)) [1, 3]. In the forward diffusion process, each step adds noise to the data, eventually making the data follow a Gaussian distribution. To generate an image $x_0$ from a Gaussian noise $x_T \sim N(0, I)$, diffusion models reverse the forward process and sample latent variables $x_t$ from a learned conditional distribution $p_\theta(x_{t-1}|x_t)$, which approximates the true reverse conditional distribution as follows:

$$p_\theta(x_{t-1}|x_t) = N(x_{t-1}; \mu_\theta(x_t, t), \sigma_t^2 I) \tag{1}$$

More specifically, by combining Eq. 1 and DDIM sampling method [6], the reverse diffusion process yields the following equation to generate a sample:

$$x_{t-1} = \sqrt{\alpha_{t-1}}\left(\frac{x_t - \sqrt{1-\alpha_t}\epsilon_\theta(x_t, t)}{\sqrt{\alpha_t}}\right) + \sqrt{1 - \alpha_{t-1} - \sigma_t^2} \cdot \epsilon_\theta(x_t, t) + \sigma_t \epsilon_t \tag{2}$$

where $\epsilon_t \sim N(0, I)$ is Gaussian noise independent of $x_t$. Here, $\alpha_t$ is a coefficient determined by the noise scheduling policy of the diffusion process, and $\sigma_t$ is a hyperparameter that controls the degree of stochasticity of the diffusion process. Note that $x_t$, the output of the diffusion step $t$, becomes the input of the next diffusion step $t - 1$, and this process continues until the image $x_0$ is reconstructed from the noise $x_T$.

$\epsilon_\theta(x_t, t)$ in Eq. 2 represents a trainable denoising network of a diffusion model. Diffusion models often use the U-Net architecture (Fig. 1(b)) for their denoising network [9]. The U-Net consists of an encoder-decoder structure in which the encoder progressively decreases the spatial dimensionality of the input image, extracting high-level features. Conversely, the decoder performs upsampling operations on the feature maps and incorporates skip connections from the encoder to reconstruct the original image resolution while generating low-level features. In the context of diffusion models,

the U-Net denoising model is trained to reduce noise in the latent variables at each diffusion step. By iteratively applying the denoising model across multiple diffusion steps, the generated images gradually exhibit improved clarity. However, it is worthwhile to note that the compute-intensive nature of the denoising network makes the generation process of diffusion models very time-consuming.

## 2.2 Post-Training Quantization

A typical uniform quantization function for converting floating-point value $x$ into integer value $x_{int}$ is as follows:

$$x_{int} = s \cdot clamp(\lfloor \frac{x - z}{s} \rceil, \ min_{quant}, \ max_{quant}) \tag{3}$$

where $s$ is a scaling factor and $z$ is the zero point. Post-training quantization (PTQ) [10, 12, 17, 18] refers to a technique for compressing neural networks that is typically applied after the network has been trained unlike quantization-aware training (QAT) [11, 19, 20, 21, 22], which considers quantization while training the network. One common PTQ strategy is to collect calibration data from the training data and calibrate the scaling factor or the rounding scheme (e.g. AdaRound [10]).

Recent studies, PTQ4DM [15] and Q-diffusion [16], have proposed PTQ methodologies for denoising network of diffusion models. PTQ4DM and Q-Diffusion provided insights into the methodology for constructing the PTQ calibration dataset for diffusion models. Specifically, since the denoising network is applied across multiple diffusion steps, these papers discussed how to consider the diffusion steps when constructing the calibration data. Both works intensively analyzed the distribution of activations in different diffusion steps, but they differ in the approaches to constructing the calibration dataset. To improve the quality of the calibration dataset, PTQ4DM adopts a method of extracting the diffusion steps from a distribution $N(\mu, 0.5T)$ where $\mu \leq 0.5T$, drawing more samples from the diffusion steps closer to $0.0T$. On the other hand, Q-diffusion divides the diffusion steps evenly into constant intervals to construct the calibration set, leveraging the similarity of activation distributions of adjacent diffusion steps. As a result, PTQ4DM achieved INT8 weights and INT8 activations for high-quality quantized diffusion models, while Q-diffusion models demonstrated INT4 weights and INT8 activations.

Previous works on quantizing diffusion models demonstrate that optimizing the construction of the calibration set can successfully reduce the bit precision of weights to as low as 4 bits [16]. However, they still require 8-bit activations, which limits network efficiency. To further reduce the activation precision, we first investigate the robustness of each diffusion step in the reverse diffusion process.

# 3 Early-Stage Robustness of Reverse Diffusion Process

## 3.1 Properties of Reverse Diffusion Process

The reverse diffusion process generates images progressively starting from Gaussian noise [8]. As shown in Fig. 2(a), the early stage of the reverse diffusion process captures the outlines of the images. In this stage, sampled images are rough and blurry sketches, so the shapes of the objects are hardly recognizable. In the later stage, because the overall structure of the images are properly shaped, the reverse diffusion process draws details of the images to improve the quality of the generation.

To quantitatively analyze the characteristics of the generated latent variables $x_t$ at each diffusion step of the reverse diffusion process, we use entropy as a measure of the randomness exhibited by the output at each step. Note that a higher entropy indicates higher randomness in the pixel values of the images, meaning they are blurrier. The entropy values of the latent variables $x_t$ at each step are summarized in Fig. 2(b). The early steps ranging from $1.0T$ to $0.5T$ maintain high entropy values, indicating that sampled images are noisy with high randomness. On the other hand, the entropy rapidly decreases after $0.5T$ step. These latter steps generate details of the image to enhance the quality of image generation, so the sampled images become sharper and their randomness decreases.

Since the characteristics of the generated images vary at each step of the reverse diffusion process, we can expect that the computational requirements for each step may differ as well. Specifically, the entropy of the sampled images remains constant with relatively high value in the early stage, suggesting that the computations performed during this stage may not need to be as precise as that for the later stages. However, in later stages where the computations have a greater impact on the entropy results, more accurate computations are necessary.

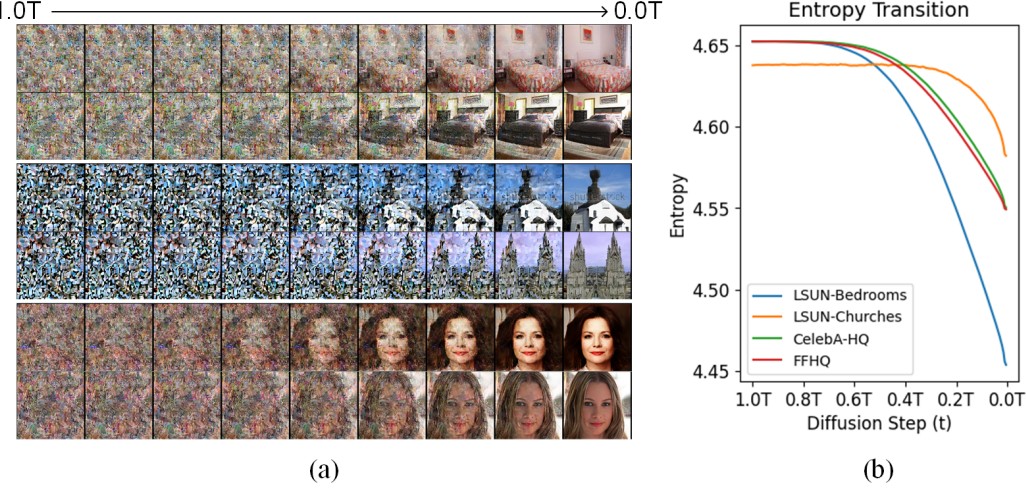

(a)                                                                (b)

Figure 2: Studies on the reverse diffusion process with respect to diffusion steps. (**a**) Examples of 256×256 LSUN-Bedrooms, LSUN-Churches, and CelebA-HQ generated in each step of the reverse diffusion process. (**b**) Entropy transition across the diffusion steps for various image generation tasks.

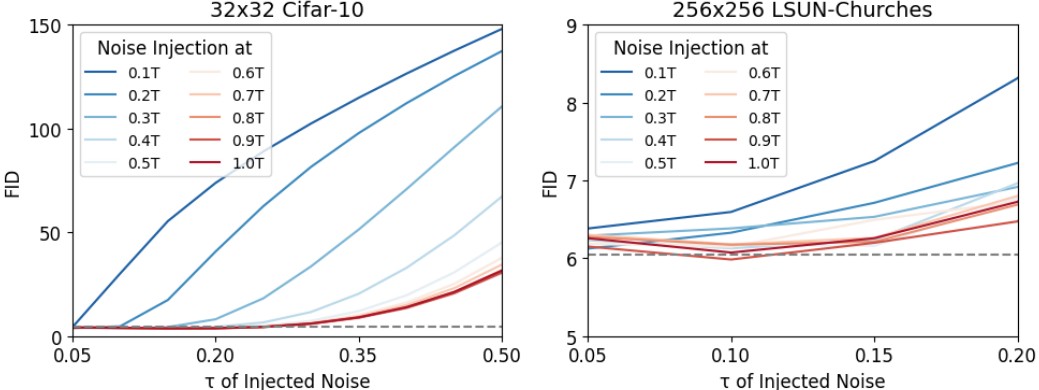

Figure 3: Image generation quality (FID) after noise injection. 5k images of (**left**) 32×32 CIFAR-10 and (**right**) 256×256 LSUN-Churches are generated for the evaluation. The magnitude of the noise is controlled by $\tau$ using Eq. 4. The diffusion step exhibits lower FID after noise injection indicates a higher level of resilience to inaccurate computations.

## 3.2   Early-Stage Robustness of Reverse Diffusion Process

In this section, we evaluate the robustness of the reverse diffusion process at each diffusion step. To measure the robustness, we inject random noise into $x_t$, the output of the target diffusion step $t$ (Eq. 2), and assess the quality of the generated image $x_0$ after the entire reverse diffusion process using the Fréchet Inception Distance (FID) [23]. The FID helps us to understand how much the computational inaccuracy at step $t$ affects the quality of overall diffusion process. The injected noise follows a Gaussian distribution $N(0, \sigma_{x_t} \cdot \tau)$, where $\sigma_{x_t}$ is the standard deviation of $x_t$, and $\tau$ controls the noise standard deviation. The noise injection modifies the latent variable $x_t$ as follows:

$$x_{t,noisy} = x_t + \epsilon, \quad \epsilon \sim N(0, (\sigma_{x_t} \cdot \tau)^2) \tag{4}$$

The analysis results are summarized in Fig. 3. The cases of 32×32 CIFAR-10 and 256×256 LSUN-Churches generation exhibit different responses to the noise injection. This discrepancy is attributed to the higher resolution of LSUN-Churches images, which introduces finer details into the generation process. Nevertheless, in both cases, there is a consistent pattern: as the standard deviation of injected noise ($\tau$) increases, the FID also increases, while the correlation between noise and FID varies signficantly depending on the diffusion step. In the early diffusion stages, close to $1.0T$, we observe

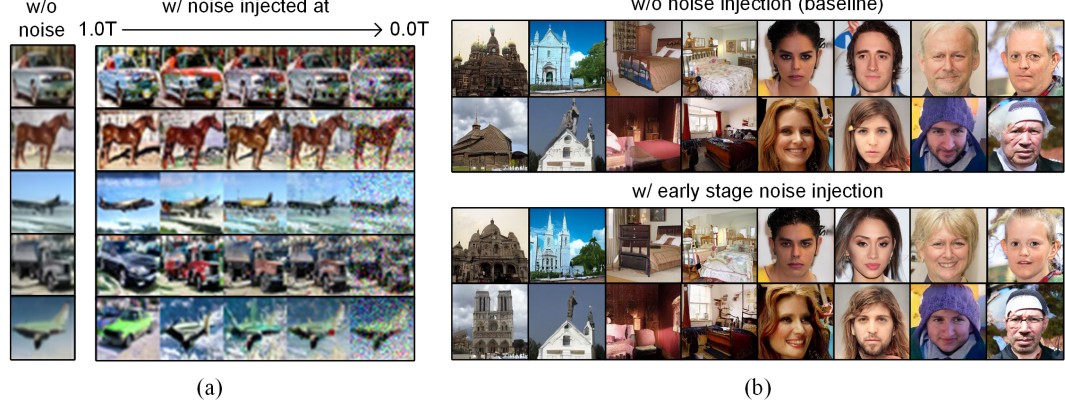

w/o noise   w/ noise injected at
1.0T ────────────→ 0.0T

w/o noise injection (baseline)

w/ early stage noise injection

(a)                                    (b)

Figure 4: Examples of image generation after noise injection. (**a**) Comparison of the CIFAR-10 images generated without noise and with noise injected in different diffusion steps. (**b**) Comparison of the generated images with and without early-stage noise injection. The early-stage noise injection changes the overall structure of the images rather than their quality.

that the quality of the image is relatively tolerant to noise. The early stages exhibit resilience to high magnitude of noise, as can be found by the maintained FID even with high $\tau$ values. However, the later stages are more susceptible to the injected noise, leading to a degradation in the FID score. This analysis confirms that the early stages are more robust to the noise than the latter stages.

To gain a more intuitive understanding of this phenomenon, we examine how the final output images change when noise is injected at each diffusion step (Fig. 4). When noise is added to the back end of the diffusion steps to capture image details, the resulting image maintains a similar overall structure but exhibits lower quality due to the injected noise. In contrast, when noise is added in the early stages, high-resolution image details remain intact while the shape of the image changes due to the noise which affects the process of capturing the overall outline. For example, as shown in Fig. 4(a) for CIFAR-10 [24] image generation, injecting noise in the initial stage causes the image at the bottom row (originally a green airplane) to become a green automobile or a slightly different shaped airplane. The diffusion model is trained to generate images similar to those belonging to the CIFAR-10 dataset, regardless of the specific class. Since both airplane and automobiles fall within the CIFAR-10 class categories, the diffusion model generates images as intended. Similarly, in the case of LSUN-bedrooms, LSUN-churches [25], CelebA-HQ [26], and FFHQ [27] generation (Fig. 4(b)), injecting noise in the early stage leads to changes in the overall image shape, rather than impacting image quality significantly. The effect of shape alteration while maintaining image quality is similar to that of modifying the random seed $x_T$ of the diffusion. The diffusion model initiates image generation from $x_T$ and gradually acquires overall shapes and colors of the images. Therefore, modification in the early stage would introduce changes to the overall structure of the generated images, not the quality.

This finding highlights the importance of understanding the varying robustness of diffusion models across different sampling steps, which can be a critical factor in improving the efficiency of denoising networks. When trying to speed up a neural network computations, approximated operations are often employed. For instance, quantization techniques approximate FP operations as integer operations to enhance processing efficiently. However, the fact that robustness varies across each step suggests that the degree to which operations can be approximated also differs across each step. Therefore, to improve efficiency without compromising the generation quality of the diffusion model, there is a great need to analyze and optimize each step individually.

## 4   Robustness-Aware Quantization of Diffusion Models

### 4.1   Impact of Weight and Activation Quantization on Diffusion Models

As the first step of quantizing the diffusion model, we evaluate the impact of quantization on each component of the denoising network. We begin by examining the impact of weight and activation

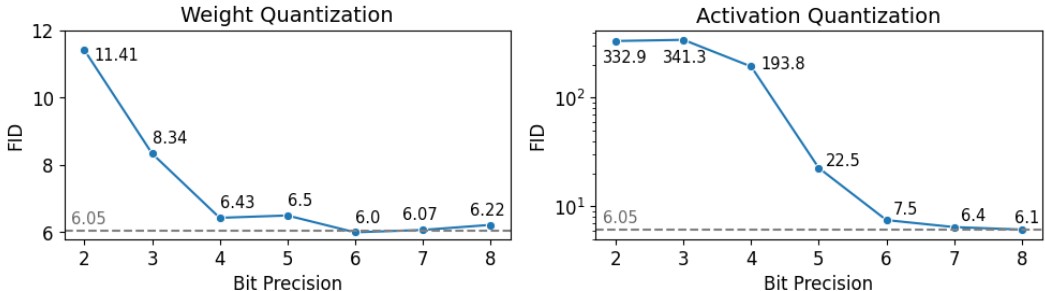

Figure 5: Image generation quality as measured by FID after (**left**) weight quantization (**right**) activation quantization of the denoising network. The diffusion model LDM-8 [8] is used to generate 5k images of 256×256 LSUN-Churches for the evaluation. The dashed gray lines indicate FID with the full-precision network. Note that the y-axis of the left graph is on a linear scale while the y-axis of the right graph is on a logarithmic scale.

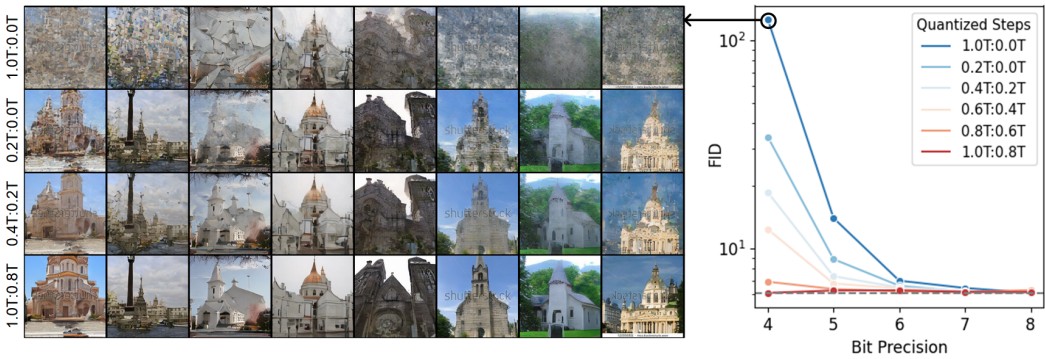

Figure 6: (**left**) Examples and (**right**) FID of 256×256 LSUN-Churches generation with activation quantization across different diffusion steps. Example images are generated by applying 4-bit activation quantization in the target diffusion steps, and FID is measured after 5k image generation. The dashed gray lines indicate FID for the full-precision network

quantization. We use BRECQ [18], a commonly utilized PTQ framework, for weight quantization and the calibration dataset is constructed based on the method proposed in Q-diffusion [16]. For activation quantization, we employ the uniform quantization function described in Eq. 3. Similar to Q-diffusion, we apply quantization for all layers of the denoising network that perform matrix multiplication, including convolution, linear, and attention layers, while keeping the activation values of attention score matrix to 8 bits. Fig. 5 shows that FID degradation is negligible up to 4 bit weight values so that low-bit weight quantization is relatively straightforward. In contrast, FID deteriorates more rapidly when the activation quantization bit decreases. Hence, we focus on reducing the number of activation bits.

### 4.2 Dynamic Nature of Activation Quantization across Diffusion Steps

Activation quantization introduces error when converting floating-point activations to integer values. In the case of uniform quantization, where a finite range of data is evenly divided using a specified number of bits, the quantization error of the target data is inversely proportional to the number of quantization bits. Furthermore, as discussed in Section 3.2, the robustness of diffusion models varies across different diffusion steps. Hence, we can expect that the bit precision of activations at each diffusion step will have distinct effects on FID. To investigate the impact of activation quantization on different diffusion steps, we divide the diffusion steps into five intervals $[1.0T{:}0.8T, 0.8T{:}0.6T,$ ..., $0.2T{:}0.0T]$, and selectively apply activation quantization to the target step interval while keeping the floating-point activation values in the other steps. For instance, if the target interval is set as $1.0T{:}0.8T$, activation quantization is applied within this interval only, while activations in the other diffusion steps remain unquantized. Fig. 6 indicate that the early-stage diffusion steps closer to $1.0T$

exhibit higher resilience to quantization errors. In the case of applying 4-bit activation quantization to the $0.2T$:$0.0T$ diffusion steps in 5k LSUN-Churches generation, the resulting FID is 34.30. However, when the same 4-bit activation quantization is applied to the $1.0T$:$0.8T$ steps, the resulting FID is 6.10, which is close to the FID of 6.05 achieved through image generation using a full-precision network.

It is worthwhile to emphasize that the dynamic nature of activation quantization during network processing eliminates the requirement of using identical activation bits for each diffusion step. This characteristic alleviates burden on adjustment of network parameters or other factors to reduce the activation bits. Consequently, by leveraging the robustness observed in the early stages, it becomes possible to further reduce activation bits, leading to more efficient processing of the denoising network.

### 4.3   Proposed Robustness-Aware Quantization

The previous section has shown that using low activation bits (e.g. 4 bits) for the denoising network in the early diffusion steps does not have a significant impact on the quality of image generation. However, as the reverse diffusion process progresses, it becomes increasingly challenging to lower the activation bits while maintaining the generated image quality. Therefore, we propose a quantization technique that leverages robustness observed in the early stages to minimize the activation bits as much as possible.

---

**Algorithm 1** Robustness-aware Quantization

---

**Require:** Weight-quantized denoising network $\epsilon_{\theta_q}$
**Require:** Lower bound of the activation bit $a$
**Require:** Number of sampled images $n$ for evaluation and threshold of FID $\eta_{fid}$
    $Bit_{act} = a$, $D_{act} = \{\}$                   $\triangleright$ $D_{act}$: Dictionary for matching steps and activation bits
    **for** $t = T$ to 1 **do**                              $\triangleright$ Optimizing activation bits for each step
        **while** ($t$ not in $D_{act}$.keys()) or ($D_{act}[t] \neq Bit_{act}$) **do**
            $D_{act}[t] = Bit_{act}$
            Sample $n$ images with $\epsilon_{\theta_q}$ applying $D_{act}$ for acitvations
            **if** FID of sampled images $> \eta_{fid}$ **then**
                $Bit_{act} = Bit_{act} + 1$
            **end if**
        **end while**
    **end for**

---

The proposed robustness-aware quantization (RAQ) is presented in Algorithm 1. It begins with pre-quantizing the weights of the denoising network, utilizing 4 bits for weight quantization, as weights are relatively easier to quantize. The optimization of activation bits starts from diffusion step $T$ and proceeds sequentially for each step. At each step, the dictionary $D_{act}$ is updated to establish the correspondence between the given activation bit $Bit_{act}$ and the diffusion step $t$. This dictionary is used by the denoising network to adjust activation bits. FP32 activations are employed for diffusion steps that have not yet been updated in the dictionary. Subsequently, using the updated bit information, the diffusion model generates $n$ small subset images, and their FID is evaluated. If the FID exceeds the desired threshold, the activation bit $Bit_{act}$ is increased by 1, and the image quality is re-evaluated. Note that RAQ employs the FID of diffusion models without activation quantization as the FID threshold to preserve the quality of image sampling after the quantization. Nonetheless, there exists the flexibility to modify the FID threshold according to the user-defined level of FID tolerance, enabling greater reduction in activation bitwidths. The iterative process continues until the desired FID is achieved. Once the FID meets the desired criterion, the optimization proceeds to the next diffusion step. Notably, the optimization starts from the previously adopted activation bit and does not explore lower bit values. This strategy acknowledges the increasing difficulty of lowering activation bits as diffusion steps progress. As a result, the time complexity for optimizing activation bits at each step is reduced from $O(m^T)$ to $O(m + T)$, where $m$ represents the number of bit candidates for activation.

Table 1: Unconditional image generation results of the baselines and the proposed RAQ. (W: weight bidwidth, A: effective activation bitwidth, TBOPs: Tera Bit Operations)

| Baseline | | | | | | | |
|---|---|---|---|---|---|---|---|
| LSUN-Churches (256x256) | | | | LSUN-Bedrooms (256x256) | | | |
| Model | W/A | FID↓ | TBOPs | Model | W/A | FID↓ | TBOPs |
| LDM-8 [8] | 32/32 | 4.09 | 4285.2 | LDM-4 [8] | 32/32 | 2.96 | 20725.8 |
| Q-diffusion [16] | 4/32 **4/8** | 4.46 **4.45** | 769.2 **148.6** | Q-diffusion [16] | 4/32 **4/8** | 4.13 **4.17** | 3120.7 **681.2** |

| Proposed RAQ (w/ 4-bit weights) | | | | | | | |
|---|---|---|---|---|---|---|---|
| LSUN-Churches (256x256) | | | | LSUN-Bedrooms (256x256) | | | |
| A | $D_{act}$ | FID↓ | TBOPs | A | $D_{act}$ | FID↓ | TBOPs |
| 6.00 | entire range - 6b | 5.31 | 108.8 | 6.00 | entire range - 6b | 5.58 | 504.8 |
| **6.00** | $[1.00T{:}0.80T]$ - 4b $[0.80T{:}0.60T]$ - 5b $[0.60T{:}0.50T]$ - 6b $[0.50T{:}0.10T]$ - 7b $[0.10T{:}0.00T]$ - 8b | **4.64** | **108.8** | **6.00** | $[1.00T{:}0.90T]$ - 4b $[0.90T{:}0.60T]$ - 5b $[0.60T{:}0.40T]$ - 6b $[0.40T{:}0.10T]$ - 7b $[0.10T{:}0.00T]$ - 8b | **3.99** | **504.8** |
| 5.60 | $[1.00T{:}0.75T]$ - 4b $[0.75T{:}0.60T]$ - 5b $[0.60T{:}0.25T]$ - 6b $[0.25T{:}0.00T]$ - 7b | 5.12 | 101.3 | 5.50 | $[1.00T{:}0.90T]$ - 4b $[0.90T{:}0.60T]$ - 5b $[0.60T{:}0.00T]$ - 6b | 4.94 | 461.6 |
| 5.35 | $[1.00T{:}0.75T]$ - 4b $[0.75T{:}0.60T]$ - 5b $[0.60T{:}0.00T]$ - 6b | 5.51 | 96.4 | 5.45 | $[1.00T{:}0.85T]$ - 4b $[0.85T{:}0.60T]$ - 5b $[0.60T{:}0.00T]$ - 6b | 5.32 | 457.3 |

## 5 Experiment

### 5.1 Unconditional Image Generation with Proposed Robustness-Aware Quantization

We first evaluate the effectiveness of the proposed method for unconditional image generation. We use Latent Diffusion Model (LDM) [8] with DDIM sampling [6] for 256×256 image generations, with a total of 200 diffusion steps. The lower bound of the activation bit $a$ is set to 4, as we observed that using 3 bits significantly degrades the quality of the generated images even with the proposed RAQ. As we find that consecutive timesteps within a 0.05T range exhibit similar sensitivity to noise injection, we set the granularity of $Bit_{act}$ update as 0.05T. We sample 5,000 images for the optimization of the quantization bits, but we use a larger sample size of 50,000 images for the final evaluation. We evaluate the performance of our approach using two datasets: 256x256 LSUN Bedrooms and LSUN Churches. Note that the generation of LSUN Bedrooms use a diffusion model LDM-4 and LSUN Churches use a diffusion model LDM-8 [8]. The quality of the generated images is evaluated using the FID metric. To estimate the efficiency improvement of RAQ, we introduce the concept of Bit Operations (BOPs), where BOPs are calculated as the product of OPs (Operations) with weight bitwidth and activation bitwidth.

The evaluation results of the baseline method, which does not employ the step-wise bitwidth optimization, are presented in the top of Table 1. Despite optimizing the activation bitwidth with a 0.05T granularity, certain consecutive intervals exhibit similar sensitivity to quantization, resulting in the the RAQ results being composed of only 5 or fewer intervals. As demonstrated in Q-diffusion [16], utilizing 4-bit weights and 8-bit activations preserves FID at a similar level to that of floating-point models. However, it is observed that FID substantially increases when 6-bit activation values are used in the entire range of the steps. In contrast, when applying our proposed RAQ method, we observe FID scores comparable to those obtained with 8-bit activations even when the effective bit count of the activation is 6 (Table 1). Thus, for LSUN-Churches/Bedrooms, the proposed RAQ

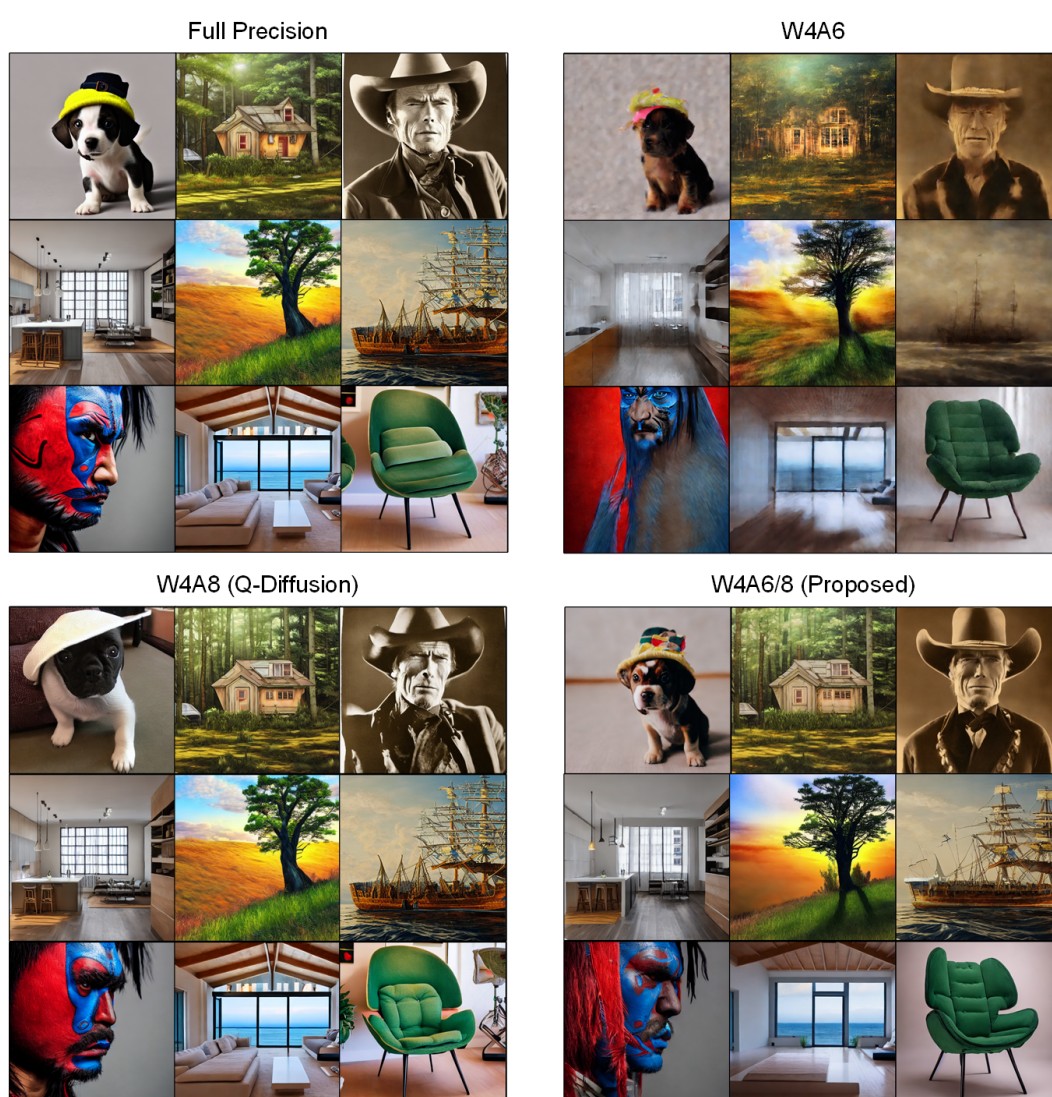

Figure 7: Text-guided $512 \times 512$ image generation results with Stable Diffusion. The denoising model with full-precision, 8-bit (Q-diffusion), and 6-bit activations, and scalable-bit acitvations with the proposed RAQ method are evaluated for the comparison. Here, W4 denotes adopting 4-bit weights, A$n$ denotes $n$-bit activations, and A$n/m$ indicates adopting $n$-bit activations in the early stage of the diffusion stages, while the later stage adopts $m$-bit activations

achieves 39.4/41.1 times smaller BOPs compared to the full-precision baseline, while Q-diffusion only achieves 28.8/30.4 times smaller BOPs compared to the full-precision baseline. Furthermore, by slightly compromising FID, it becomes possible to further reduce the effective activation bits. These experimental results clearly demonstrate that leveraging the early-stage robustness of the diffusion model contributes to maintaining the quality of image generation even with low-bit activations.

## 5.2   Conditional Image Generation with Proposed Robustness-Aware Quantization

In this section, we assess the effectiveness of the proposed method for conditional image generation using Stable Diffusion [8]. We employ the PLMS sampler [28] with 50 diffusion steps to generate 512x512 images. Since there is no established metric to quantitatively measure the image generation quality in this case, we rely on qualitative comparison for each image. As presented in Figure 7, even when the proposed RAQ method is not applied, 8-bit activations produce high-quality images that are comparable with full-precision models. However, there is a significant degradation in image quality

when the activation bits are reduced to 6 bits. On the other hand, by applying 6-bit activations only to the $[1.0T{:}0.8T]$ diffusion steps and retaining 8-bit activations for the remaining steps, the diffusion models are able to generate high-quality images. This result underscores the importance of leveraging the early-stage robustness of the diffusion process.

Meanwhile, unlike the unconditional image generation cases in Section 5.1, we had difficulties in applying 4-bit activations to the early-stage diffusion process of Stable Diffusion. This challenge arises from the large dynamic range of the activation values in Stable Diffusion. Hence, we plan to explore more sophisticated techniques to address this issue in the future work.

# 6  Limitation

An important limitation of the proposed RAQ is that current results do not enhance running time efficiency on GPUs. This is primarily due to the substantial presence of irregular activation bitwidths, such as 6 bits (Table 1), and accelerating diffusion models with irregular bitwidths on a GPU does not lead to performance improvement due to the lack of corresponding arithmetic units in GPU.

However, we would like to emphasize that our main contribution is to show a direction that we can reduce bit resolution for a part of parameters, and our approach is not fundamentally limited to a certain bit resolution. Hence, if other works that can further reduce the overall bit resolution of the network are developed independent of our scheme, then it can be combined with our proposed scheme, and there is a chance that portion of the 4-bit parameters in our scheme can be increased to see realistic performance benefits on GPU. For example, our current approach involves a basic min/max-based quantization mechanism for activation quantization, leading to an 8-bit quantization for the baseline fixed bitwidth case. Then, the allocation of activation bits using the RAQ method is distributed as (4b, 5b, 6b, 7b, 8b) = (20%, 20%, 10%, 40%, 10%) for LSUN-churches (Table 1). However, in the scenario where an advanced quantization mechanism enables 6-bit activation quantization even for the baseline fixed bitwidth case, there exists the possibility of increasing the proportion of 4-bit activations with the proposed RAQ, such as (4b, 5b, 6b) = (50%, 30%, 20%), which can then be effectively accelerated on GPUs.

Moreover, the utilization of specialized hardware, like bit-scalable accelerators, could offer a promising solution for processing these models [29]. These accelerators are purpose-built to harness the benefits of quantization on a bit-by-bit basis, resulting in a nearly linear improvement in computing efficiency, encompassing both latency and energy consumption, with decreasing bit width. To show potential benefit from such bit-scalable hardware, we analyze the BOPs in Table 1. This metric allows us to estimate the performance gain achievable through the RAQ method. In the case of LSUN-Churches, the full-precision baseline and Q-diffusion necessitates 4285.2 and 148.6 TBOPs for a single image generation respectively, while the proposed RAQ method only requires 108.8 TBOPs. This indicates that with the utilization of specialized accelerators, the implementation of the RAQ approach could potentially lead to more than 39.4 times speedup and energy savings compared to the full-precision baseline, while Q-diffusion can achieve 28.8 times improvement compared to the full-precision baseline.

# 7  Conclusion

In this paper, we present a comprehensive analysis of the reverse diffusion process and introduce a strategy to optimize diffusion models using quantization techniques. Leveraging the observed robustness of the early stages in the diffusion process, we successfully optimize the activation bitwidth for each diffusion step, leading to reduced effective activation bit precision without compromising the fidelity and diversity of the generated images. Experimental results show that our proposed method (RAQ) effectively reduces the effective bitwidth of activations to 6 bits while maintaining image quality comparable to conventional approaches that employ 8-bit activations. Additionally, we also evaluate the effectiveness of the proposed method in the context of conditional image generation using Stable Diffusion, a state-of-the-art diffusion model.

## Acknowledgements

This work was supported in part by Institute of Information & communications Technology Planning & Evaluation (IITP) grant funded by the Korea government (MSIT) (No. 2021-0-00105: Development of model compression framework for scalable on-device AI computing on Edge applications , IITP-2023-RS-2023-00256081: artificial intelligence semiconductor support program to nurture the best talents, No. 2021-0-01343: Artificial Intelligence Graduate School Program (Seoul National University), No.2021-0-02068: Artificial Intelligence Innovation Hub), Basic Science Research Program through the National Research Foundation of Korea (NRF) funded by the Ministry of Education (2022R1A6A3A01087416), BK21 FOUR program and Inter-university Semiconductor Research Center at Seoul National University. (Corresponding Author: Jae-Joon Kim)

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

# A Experimental Details

## A.1 Details of Diffusion Models

In order to conduct a comprehensive analysis of the early-stage robustness in diffusion models, we employ five different diffusion models to generate various datasets, including CIFAR10 [24], FFHQ [27], CelebA-HQ [26], LSUN-bedrooms, and LSUN-churches [25]. To meausre the Fréchet Inception Distance (FID), we utilize the torch-fidelity library [1], following the methodology established in previous works [6, 8]. Subsequently, we apply the proposed RAQ method to perform high-resolution image generation tasks, encompassing both unconditional image generation for LSUN-bedrooms/LSUN-churches and conditional image generation using Stable Diffusion [8]. The details of the diffusion models utilized in these experiments are thoroughly presented in Table 2. Note that the calculation of $\sigma_t$ in Eq. 2 is performed using $\eta$ in Table 2 as specified in the following equation [6]:

$$\sigma_t = \eta \cdot \sqrt{(1 - \alpha_{t-1})/(1 - \alpha_t)}\sqrt{1 - \alpha_t/\alpha_{t-1}} \tag{5}$$

Table 2: Implementation specifications of the diffusion models

|  | CIFAR-10 | FFHQ/ CelebA-HQ | LSUN- Bedrooms | LSUN- Churches | Conditional Generation |
|---|---|---|---|---|---|
| Image Size | 32×32 | 256×256 | 256×256 | 256×256 | 512×512 |
| Architecture | DDIM [2] | LDM-4 [3] | LDM-4 [3] | LDM-8 [3] | Stable Diffusion v1.4 [4] |
| Sampler | DDIM [6] | DDIM | DDIM | DDIM | PLMS [28] |
| Step Count | 100 | 200 | 200 | 200 | 50 |
| $\eta$ | 0 | 0/1 | 1 | 0 | 0 |

## A.2 Details of Entropy Analysis

In Section 3.1, we calculate the entropy of the latent variables $x_t$ for each diffusion step. To facilitate this calculation, we transform the values of $x_t$ into histogram bins. Specifically, we map $x_t$ to a histogram bin using the following equation:

$$h(x_t) = clamp(\lfloor \frac{x_t}{256} \rceil, -3, +3) \tag{6}$$

This equation ensures that the values of the histogram bins are constrained within the range of $-3$ to $+3$, allowing us to effectively create a histogram with 256 bins. Once we have the histogram representation of $x_t$, we can calculate the entropy using the equation[5]:

$$H(X) = \sum_x -p(X) \log p(X) \tag{7}$$

## A.3 Intermediate Image Prediction during Reverse Diffusion Process

In Fig. 1(a) and Fig. 2, we showcase the intermediate image prediction results during the reverse diffusion process, aiming to illustrate the characteristics of diffusion models. To visualize the prediction results, we utilize the following $x_0$ prediction of each diffusion step as stated in [6]:

$$x_0 = \frac{x_t - \sqrt{1 - \alpha_t}\epsilon_\theta(x_t, t)}{\sqrt{\alpha_t}} \tag{8}$$

---

[1] https://github.com/toshas/torch-fidelity
[2] https://github.com/ermongroup/ddim
[3] https://github.com/CompVis/latent-diffusion
[4] https://github.com/CompVis/stable-diffusion
[5] Claude Elwood Shannon, "A Mathematical Theory of Communication", Bell system technical journal, 1948

### A.4 Details of Activation Quantization

During the activation quantization process, we observed that the skip connections of ResBlocks, the first convolutional layer responsible for transforming the latent variable into the input of the denoising network, and the last convolutional layer responsible for transforming the output of the denoising network (Fig. 1(b)) had a significant impact on the quality of the final image. However, these components constitute a negligible fraction of the overall computation. Therefore, in order to balance computational efficiency and image quality, the proposed RAQ adjust the activation bits of the diffusion models to the desired bit precision while keeping the activation bits of these three components fixed at 8 bits.

## B Additional Results

### B.1 Activation Quantization across Diffusion Steps

Fig. 9 complements the image generation results presented in Fig. 6. Fig. 9 showcases image generation with 4-bit activation quantization at different diffusion steps, alongside the image generation with floating-point activations. The results of the activation quantization demonstrate a consistent trend with the noise injection test (4). When 4-bit activation quantization is applied to the early stages, the resulting images closely resemble those generated using floating-point activations, showcasing high quality with minor shape variations. However, applying 4-bit activation quantization to the entire diffusion process leads to a significant compromise in the generated image quality. This is primarily due to the degradation in image quality caused by the quantization applied to the later diffusion steps.

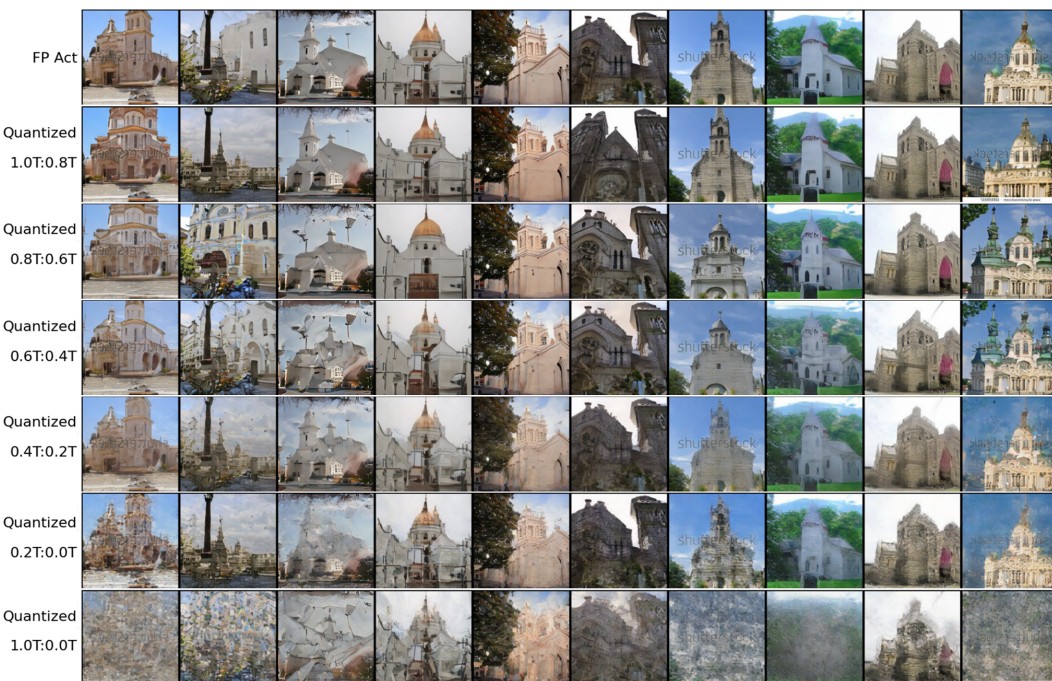

Figure 8: Examples of 256×256 LSUN-Churches generation with FP32 activation or activation quantization across different diffusion steps. Example images with activation quantization are generated by applying 4-bit activation quantization in the target diffusion steps.

### B.2 Explanation on FID Improvement with Proposed RAQ on LSUN-Bedrooms

In Table 1 of Section 5.1, we observe that the LSUN-Bedrooms images generated with the proposed RAQ exhibit slightly better FID scores compared to Q-diffusion with W4A32 and W4A8. To investigate the reason behind this FID improvement, we conduct a detailed comparison of the images generated using full-precision activations and 4-bit activations in the early stage of the diffusion

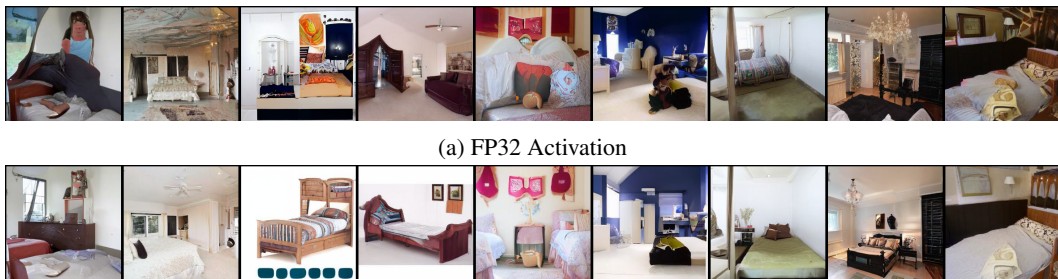

(a) FP32 Activation

(b) 4-bit Activation for $[1.0T{:}0.8T]$

Figure 9: Examples of 256×256 LSUN-Bedrooms generation with different activation precision.

process. We find that the models with full-precision activations sometimes generate images with complex structures that are not easily recognizable as bedrooms. However, when 4-bit activation quantization is applied to the early stage, it simplifies the complex structures and results in images that more closely resemble bedrooms. This observation suggests that the step-wise activation quantization strategy employed in the proposed RAQ method helps refine the generated images, leading to improved quality and better alignment with the target LSUN-Bedrooms dataset.

## B.3 Activation Quantization of Stable Diffusion

For the conditional image generation with Stable Diffusion, we utilize the prompt examples that are publicly available online [6] [7]. The prompts used in Section 5.2 are as follows:

1. a puppy wearing a hat

2. cluttered house in the woods in anime oil painting style*

3. Old photo of Clint Eastwood dressed as cowboy,1800s, centered, by professional photographer, wide-angle lens, background saloon*

4. interior design, open plan, kitchen and living room, modular furniture with cotton textiles, wooden floor, high ceiling, large steel windows viewing a city Artstation and Antonio Jacobsen and Edward Moran, (long shot), clear blue sky, intricated details, 4k*

5. a tree on the hill, bright scene, highly detailed, realistic photo

6. a highly detailed, majestic royal tall ship on a calm sea,realistic painting, by Charles Gregory*

7. medium shot side profile portrait photo of the Takeshi Kaneshiro warrior chief, tribal panther make up, blue on red, looking away, serious eyes, 50mm portrait, photography, hard rim lighting photography –ar 2:3 –beta –upbeta

8. a picture of dimly lit living room, minimalist furniture, vaulted ceiling, huge room, floor to ceiling window with an ocean view, nighttim*

9. an armchair in the shape of an avocado, an armchair imitating an avocado*

In this section, we additionally present non-cherry-picked samples generated using Stable Diffusion with and without activation quantization. We use the prompts that are highlighted with asterisk (*). For each prompt, we generate four images to demonstrate the variety and quality of the generated results (Fig. 10 in the next page).

---

[6]https://stablediffusion.fr/prompts

[7]https://mpost.io/best-100-stable-diffusion-prompts-the-most-beautiful-ai-text-to-image-prompts/

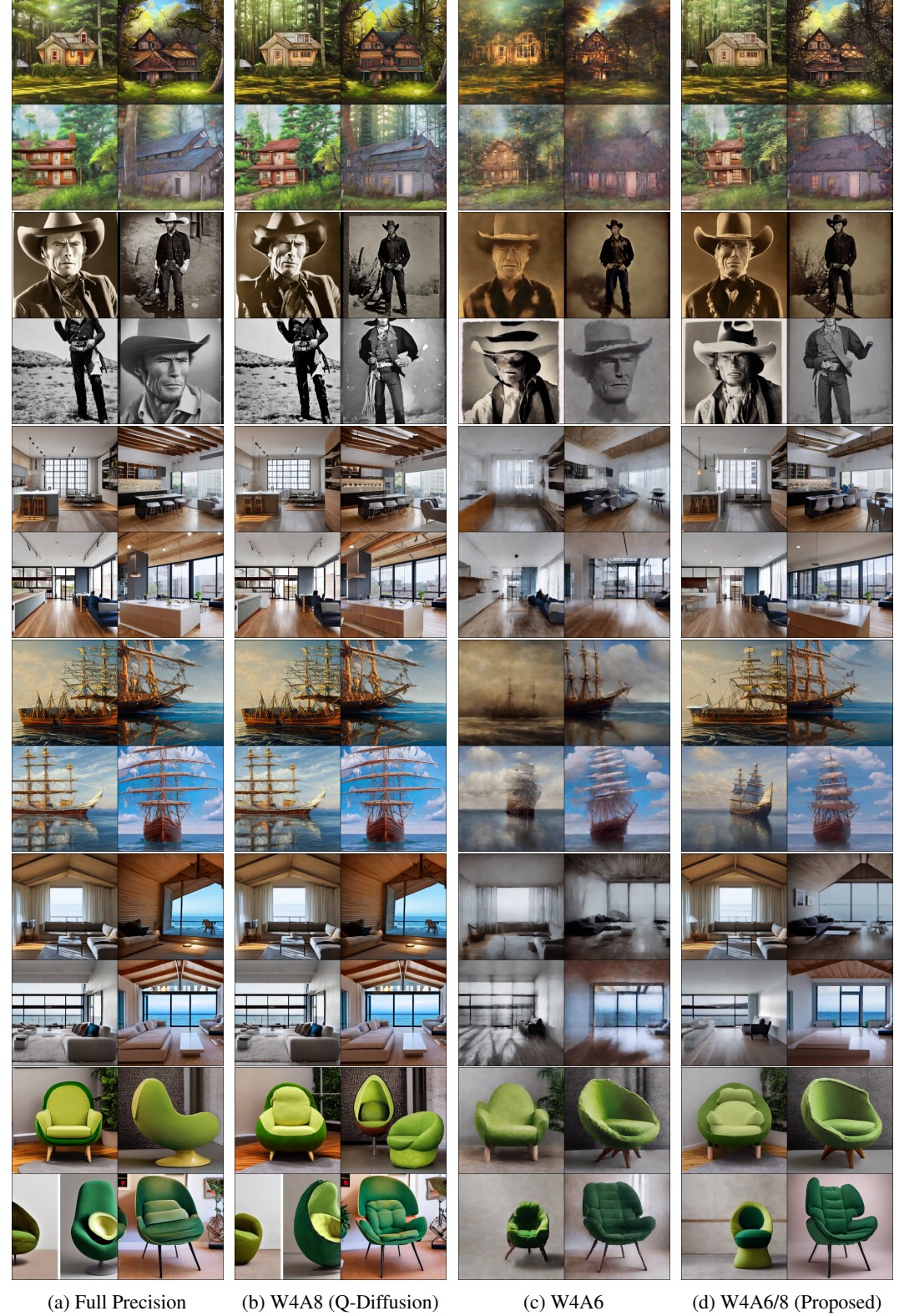

(a) Full Precision          (b) W4A8 (Q-Diffusion)          (c) W4A6          (d) W4A6/8 (Proposed)

Figure 10: Text-guided 512×512 image generation results with Stable Diffusion.

