# OpenReview forum: "Leveraging Early-Stage Robustness in Diffusion Models for Efficient and High-Quality Image Synthesis"
_NeurIPS.cc/2023/Conference — NeurIPS 2023 poster_

### Official Review · Reviewer_ctiM · 2023-06-27

**Soundness:** 4 excellent
**Presentation:** 4 excellent
**Contribution:** 3 good
**Rating:** 6
**Confidence:** 4

**Summary:**

The authors design robustness-aware quantization (RAQ) to speed up the noise estimation network by leveraging the robustness of early-stage diffusion models. Specifically, the authors found that the quality of generated images is less affected by the early-stage. Therefore, they reduce the bitwidth of activations for the early-stage, and maintain high-bit activations for the later-stage. Experiments show that the proposed method can speed up early computations while maintaining generation quality.

**Strengths:**

1. The idea of the paper is simple yet effective, promoting the application of Post-Training Quantization (PTQ) in the diffusion model.
2. The analyses in the paper are extensive. The authors demonstrate the early-stage robustness through entropy transition across steps (Fig. 2) and noise injection.
3. The paper is well-organized and easy to read.
4. The authors also provide the code for results reproduction, showing the solidness of the work.

**Weaknesses:**

1. In Tab. 1, RAQ only sets different bitwidths in five intervals, which is inconsistent with Algorithm 1, which sets different bitwidths in each step. Some explanation is needed.
2. The paper only reduces the activation bitwidth on the base of Q-diffusion. However, compared with LDM-4, the FID of Q-diffusion increased by 1.21 about LSUN-Bedrooms (256x256) in Tab. 1. Therefore, the activation bitwidth should not be reduced only. It is better to further apply RAQ to weight bidwidth to obtain a better trade-off between performance and efficiency.
3. The authors propose RAQ to accelerate the early-stage computation. However, the running time, FLOPs, and model size are not provided to demonstrate the effectiveness of the proposed method.

**Questions:**

1. Can performance and computation be further optimized with finer-grained (beyond five intervals, even one interval per step) settings?
2. In Tab. 1, the FID of RAQ (3.99) with smaller activation bitwidth is better than Q-diffusion (4.17) about LSUN-Bedrooms (256x256). Please give some analyses about these results.
3. In Fig. 7, the generation quality of the W4A6/8 has a large difference compared with W4A8 and full precision (especially the two images in the lower right corner and upper left corner). This result is different from the quantitative comparison in Tab. 1, where RAQ has comparable (even better) FIDs but smaller activation bitwidth. Please give some explanations.

**Limitations:**

The authors do not discuss its limitations or potential negative societal in separate section. It is better if the authors add some discussion.

---

> ### Author Rebuttal · Authors · 2023-08-10
>
> **Response ctiM-1: The granularity of bitwidth optimization**
>
> In the context of the RAQ method outlined in Algorithm 1, choosing a finer granularity for Bit_act update necessitates a larger number of sampled images for the optimization process. Meanwhile, our investigation revealed that consecutive timesteps within a 0.05T range exhibit similar sensitivity to noise injection. Therefore, our experimental configuration concentrated on optimizing the activation bitwidth with intervals of 0.05T for the Bit_act update. This approach ensures effective optimization while managing computational demands.
> On the other hand, despite optimizing the activation bitwidth with a granularity of 0.05T, some consecutive intervals present the same level of sensitivity to the quantization and thus share the optimal bitwidth. Therefore, the experimental results summarized in Table 1 are composed of only 5 intervals or even fewer intervals.
>
> **Response ctiM-2: Applying RAQ to weight bitwidth optimization**
>
> We agree with your opinion that, by implementing the RAQ method on weight quantization, it's possible to prevent FID increase in scenarios involving Q-diffusion with 4-bit weight quantization. However, it is important to note that applying the RAQ method on the weight quantization is not within the scope of this paper. The principal aim of this research is to enhance the diffusion inference process without introducing supplementary costs into the inference stage. This is especially achievable in the context of implementing the RAQ method on activation quantization, it does not entail any extra costs such as additional memory requirements.
>
> Conversely, applying RAQ to weight quantization increases the memory requirements due to the potential increase in the number of parameters, as we have to save copies of parameters with different resolutions.
>
> **Response ctiM-3: The effectiveness of the proposed RAQ**
>
> As the reviewer pointed out, the operation counts will be updated in the final version.
>
> We introduce the concept of Bit Operations (BOPs), where BOPs are calculated as the product of OPs with weight and activation bitwidths. This metric allows us to estimate the performance gain achievable through the RAQ method. In the case of LSUN-Churches, the full-precision baseline and Q-diffusion necessitates 4285.2 and 148.6 TBOPs (TBOPs: Tera BOPs) for a single image generation respectively, while the proposed RAQ method only requires 108.8 TBOPs. This indicates that the RAQ approach could potentially lead to more than 39.4 times speedup and energy savings compared to the full-precision baseline, while Q-diffusion can achieve 28.8 times improvement compared to the full-precision baseline.
>
> Table A-1. LSUN-Churches (256x256) generation results
> | Model |W/A | FID | TBOPs|
> |---|---|---|---|
> | LDM-8 |32/32 | 4.09 | 4285.2 |
> | Q-diffusion | 4/8 | 4.45 | 148.6 |
> | RAQ | 4/6 | 4.64 | 108.8 |
>
> Table A-2.LSUN-Bedrooms (256x256) generation results
> | Model |W/A | FID | TBOPs|
> |---|---|---|---|
> | LDM-4 |32/32 | 2.96 | 20725.8 |
> | Q-diffusion |4/8 | 4.17 | 681.2 |
> | RAQ |4/6 | 3.99 | 504.8 |
>
> **Response ctiM-4: The impact of bitwidth optimization with finger-grained settings**
>
> Thank you for the insightful question. However, further optimization is hardly achievable with finer-grained settings. As mentioned in **Response ctiM-1** regarding interval granularity, the optimization process was indeed carried out with a finer-grained approach. However, the optimization process resulted in only 5 intervals or even fewer intervals, because consecutive time steps exhibit similar sensitivity to quantization. Consequently, the potential for achieving further optimization through finer-grained settings becomes limited.
>
> **Response ctiM-5: Explanation on FID improvement with proposed RAQ on LSUN-bedrooms**
>
> As you pointed out, it is interesting that LSUN-Bedrooms images generated with the proposed RAQ exhibit slightly better FID scores compared to Q-diffusion with W4A32 and W4A8. To investigate the reason behind this FID improvement, we conduct a detailed comparison of the images generated using full-precision activations and 4-bit activations in the early stage of the diffusion process. process. We find that the models with full-precision activations sometimes generate images with complex structures that are not easily recognizable as bedrooms. However, when 4-bit activation quantization is applied to the early stage, it simplifies the complex structures and results in images that more closely resemble bedrooms. This observation suggests that the step-wise activation quantization strategy employed in the proposed RAQ method helps refine the generated images, leading to improved quality and better alignment with the target LSUN-Bedrooms dataset.
> The detailed discussion with sampled images are updated in Supplementary Material B.2.
>
>
> **Response ctiM-6: The generation quality of the W4A6/8 Stable Diffusion model**
>
> As you highlighted, there is a quality issue when generating high-resolution (512x512) images using the Stable Diffusion model with the W4A6/8 configuration.This is due to the intricate nature of capturing fine details in high-resolution images, which demands exceptionally accurate computations. Consequently, this case becomes more sensitive to quantization effects compared to the 256x256 image generation instances reported in Table 1.
>
> However, it is important to note that our experimental results consistently emphasize the effectiveness of deploying a mixed precision strategy through the RAQ method. Specifically, the mixed-precision quantization achieved by the RAQ approach (W4A6/8) significantly enhances the quality of the generated images when contrasted with scenarios where activation bits are fixed at 6 bits (W4A6).
>
> Thanks for the constructive comments.

---

> > ### Comment · Reviewer_ctiM · 2023-08-15
> >
> > Thanks for the rebuttal. I read the authors' responses. I also read the comments and rebuttals from other reviews. The authors' reply solves my concerns, including bitwidth, FID improvement, generation quality, and effectiveness of RAQ.
> >
> > But I don't understand why the authors use BOPs instead of some common metrics such as FLOPs and running time.
> >
> > And it would be better if the authors discuss the limitations of the work, which Reviewer 4DBQ also mentions.

---

> > > ### Author Response · Authors · 2023-08-15
> > > **Discussion on BOPs and the limitations of the proposed work (1/2)**
> > >
> > > Thank you for the valuable comments.
> > >
> > > As you rightly pointed out, when evaluating the efficiency of neural network models, metrics such as FLOPs and running time are commonly used. However, several prior works on quantized neural networks, such as NIPQ [1], have adopted BOPs for evaluating the efficiency of quantized models for the following two reasons.
> > >
> > > Firstly, FLOPs measure the number of floating-point operations, while the proposed RAQ involves integer operations in processing diffusion models as both weights and activations are integer. Consequently, it is challenging to compare quantized models' efficiency with FLOPs.
> > >
> > > Secondly, while running time is a reliable efficiency metric for neural network models, accelerating diffusion models quantized with irregular bit widths on GPUs is also challenging due to the absence of corresponding arithmetic units.
> > > In the context of hardware, computation efficiency is intrinsically dependent on both operation count and bitwidth, as computing units are composed of multiple binary logics. The required number of these binary logics is influenced by both the operation count and the bitwidth. Hence, there is a direct correlation between BOPs and computation efficiency like energy efficiency and latency, especially in bit-scalable accelerators.
> > >
> > > In summary, we agree that our current RAQ results has limitation in showing improvements in running time on GPUs due to significant proportion of irregular activation bitwidths. To overcome this limitation, potential directions include the utilization of bit-scalable accelerators to improve running time, or the exploration of more sophisticated quantization schemes that could increase the portion of 4-bit activations, which can then be effectively accelerated on GPUs.
> > >
> > > [1] Shin, Juncheol, et al. "NIPQ: Noise Proxy-Based Integrated Pseudo-Quantization." Proceedings of the IEEE/CVF Conference on Computer Vision and Pattern Recognition. 2023.

---

> > > > ### Author Response · Authors · 2023-08-15
> > > > **Discussion on BOPs and the limitations of the proposed work (2/2)**
> > > >
> > > > As suggested, we will update the limitation section in the final manuscript to reflect such limitations in our work as follows.
> > > >
> > > > **Limitation Section**
> > > >
> > > > An important limitation of the proposed RAQ is that current results do not enhance running time efficiency on GPUs. This is primarily due to the substantial presence of irregular activation bitwidths, such as 6 bits (Table 1), and accelerating diffusion models with irregular bitwidths on a GPU does not lead to performance improvement due to the lack of corresponding arithmetic units in GPU.
> > > >
> > > > However, we would like to emphasize that our main contribution is to show a direction that we can reduce bit resolution for a part of parameters, and our approach is not fundamentally limited to a certain bit resolution. Hence, if other works that can further reduce the overall bit resolution of the network are developed independent of our scheme, then it can be combined with our proposed scheme, and there is a chance that portion of the 4-bit parameters in our scheme can be increased to see realistic performance benefits on GPU. For example, our current approach involves a basic min/max-based quantization mechanism for activation quantization, leading to an 8-bit quantization for the baseline fixed bitwidth case. Then, the allocation of activation bits using the RAQ method is distributed as (4b, 5b, 6b, 7b, 8b) = (20%, 20%, 10%, 40%, 10%) for LSUN-churches (Table 1). However, in the scenario where an advanced quantization mechanism enables 6-bit activation quantization even for the baseline fixed bitwidth case, there exists the possibility of increasing the proportion of 4-bit activations with the proposed RAQ, such as (4b, 5b, 6b) = (50%, 30%, 20%), which can then be effectively accelerated on GPUs.
> > > >
> > > > Moreover, the utilization of specialized hardware, like bit-scalable accelerators, could offer a promising solution for processing these models [1]. These accelerators are purpose-built to harness the benefits of quantization on a bit-by-bit basis, resulting in a nearly linear improvement in computing efficiency, encompassing both latency and energy consumption, with decreasing bit width. To show potential benefit from such bit-scalable hardware, we analyze the Bit Operations (BOPs) characteristics, where BOPs are calculated as the product of OPs (Operations) with weight bitwidth and activation bitwidth. This metric allows us to estimate the performance gain achievable through the RAQ method. In the case of LSUN-Churches, the full-precision baseline and Q-diffusion necessitates 4285.2 and 148.6 TBOPs (TBOPs: Tera BOPs) for a single image generation respectively, while the proposed RAQ method only requires 108.8 TBOPs. This indicates that with the utilization of specialized accelerators, the implementation of the RAQ approach could potentially lead to more than 39.4 times speedup and energy savings compared to the full-precision baseline, while Q-diffusion can achieve 28.8 times improvement compared to the full-precision baseline.
> > > >
> > > >
> > > > [1] Fu, Yonggan, et al. "2-in-1 accelerator: Enabling random precision switch for winning both adversarial robustness and efficiency." MICRO-54: 54th Annual IEEE/ACM International Symposium on Microarchitecture. 2021.
> > > >
> > > >
> > > > Once again, thank you for your valuable feedback. We greatly appreciate your insights and remain open to continued discussion.

---

> > > > > ### Comment · Reviewer_ctiM · 2023-08-20
> > > > >
> > > > > Thanks for your response. It solves my concern. I'm happy to increase my score to 6 (weak accept).

---

### Official Review · Reviewer_4DBQ · 2023-07-03

**Soundness:** 3 good
**Presentation:** 3 good
**Contribution:** 3 good
**Rating:** 5
**Confidence:** 4

**Summary:**

This paper proposes to quantize diffusion models to a different extent along the iterative process for image generation. The main motivation of the proposed approach is that diffusion model is robust to input distortion at early stages (i.e. noisy stages) of the iterative process. Therefore, the proposed approach starts with a 4-bit quantization, and gradually increase activation bits along the iterations. Experiments show that the proposed approach achieves improved performance with the same effective bitwidth.

**Strengths:**

1. This paper has a good motivation. The empirical experiments show that it is legitimate to apply different rates of quantization at different stages of different diffusion process.

2. The proposed method effectively improves the performance with a reduced bitwidth, as shown in Table 1.

3. The idea is simple and is easy to follow.

**Weaknesses:**

1. From Table 1, the bitwidth for each timestep is model-specific. That means optimization has to be done for each model. It would be good to have analysis on the robustness of the bitwidth selection.

2.  As one of the main objectives is to improve the sampling efficiency, the comparison of runtime should be included. This is important for readers to understand the improvement brought by the proposed method.

**Questions:**

Overall, I think the dynamic quantization in this paper is a legitimate approach for improving the efficiency of diffusion model sampling. Please refer to the weakness sections for the questions.

**Limitations:**

Please consider discussing the limitations and potential societal impact of the proposed approach in the paper.

---

> ### Author Rebuttal · Authors · 2023-08-10
>
> Thanks for the constructive comment.
>
> We will now discuss 1) the bitwidth selection for different models, and 2) improvement of the sampling efficiency with RAQ.
>
> **Response 4DBQ-1: The bitwidth selection for different models**
>
> As you correctly highlighted, the bitwidth optimization with the proposed RAQ is a model-specific optimization. The experimental results presented in Table 1 have been attained through the activation bitwidth optimization process as illustrated in Algorithm 1. The optimization of bitwidth is executed iteratively for each model, involving FID measurement. However, it is worth noting, as detailed in Section 4.3, that the proposed RAQ method holds the potential to reduce the optimization time required for model-specific bitwidth optimization from $O(m^T)$ to $O(m+T)$
>
> **Response 4DBQ-2: Improvement of the sampling efficiency with RAQ**
>
> While the accelerating diffusion models quantized with irregular bit widths (e.g., 6bits) on a GPU presents challenges, the utilization of specialized hardware, like bit-scalable accelerators, could offer a promising solution for processing these models [1]. These accelerators are purpose-built to harness the benefits of quantization on a bit-by-bit basis, resulting in a nearly linear improvement in computing efficiency, encompassing both latency and energy consumption, with decreasing bit width. We introduce the concept of Bit Operations (BOPs), where BOPs are calculated as the product of OPs with weight bitwidth and activation bitwidth. This metric allows us to estimate the performance gain achievable through the RAQ method. In the case of LSUN-Churches, the full-precision baseline and Q-diffusion necessitates 4285.2 and 148.6 TBOPs (TBOPs: Tera BOPs) for a single image generation respectively, while the proposed RAQ method only requires 108.8 TBOPs. This indicates that with the utilization of specialized accelerators, the implementation of the RAQ approach could potentially lead to more than 39.4 times speedup and energy savings compared to the full-precision baseline, while Q-diffusion can achieve 28.8 times improvement compared to the full-precision baseline.
>
> Table A-1. LSUN-Churches (256x256) generation results
> | Model |W/A | FID | TBOPs|
> |---|---|---|---|
> | LDM-8 |32/32 | 4.09 | 4285.2 |
> | Q-diffusion | 4/8 | 4.45 | 148.6 |
> | RAQ | 4/6 | 4.64 | 108.8 |
>
> Table A-2.LSUN-Bedrooms (256x256) generation results
> | Model |W/A | FID | TBOPs|
> |---|---|---|---|
> | LDM-4 |32/32 | 2.96 | 20725.8 |
> | Q-diffusion |4/8 | 4.17 | 681.2 |
> | RAQ |4/6 | 3.99 | 504.8 |
>
> [1] Fu, Yonggan, et al. "2-in-1 accelerator: Enabling random precision switch for winning both adversarial robustness and efficiency." MICRO-54: 54th Annual IEEE/ACM International Symposium on Microarchitecture. 2021.

---

> > ### Comment · Reviewer_4DBQ · 2023-08-18
> >
> > Thank you for the authors' response. The response has addressed my questions and hence I would keep my current rating.

---

### Official Review · Reviewer_ZFyZ · 2023-07-07

**Soundness:** 3 good
**Presentation:** 2 fair
**Contribution:** 2 fair
**Rating:** 5
**Confidence:** 5

**Summary:**

This paper presents robustness-aware quantization (RAQ), a novel strategy to use mixed precisions for activations when quantizing diffusion models. The authors found that inaccurate computation during the early stages of the reverse diffusion process has minimal impact on the quality of generated images, and propose to use low-bit activations for the early reverse diffusion process while maintaining high-bit activations for the later stages. Experiments have been conducted for both unconditional and conditional generation using latent diffusion and stable diffusion on various datasets.

**Strengths:**

- The paper is well-structured and presents a clear motivation for leveraging the robustness of early-stage diffusion models to use lower-bit activations at those time steps to further improve the computation efficiency.
- Experimental results show that the proposed method can use lower precisions for early-stage computation without sacrificing the quality of the generated images.
- The experiments with stable diffusion indicate the effectiveness of the proposed methods on text-to-image applications.

**Weaknesses:**

My biggest concern with the proposed RAQ approach is its practicality. The method suggests using low-bit activations for the early denoising process and high-bit activations for the later stages. However, the paper does not provide sufficient arguments on how this varying precision can be efficiently implemented and how much additional benefits it can bring compared to the simple W4A8 cases. In real-world applications, changing activation precisions could introduce complexities in designing and implementing corresponding kernels for different stages of the process, as the weight precisions need to be always upcasted to the activation precisions when performing the compute on conventional GPUs (e.g. the compute will always be WyAy for WxAy precisions, where x=4 and y>=4 for the settings discussed in the paper). Consequently, this could limit the practical utility and impact of the RAQ approach. An analysis of the theoretical speed up or memory saving should be done to show that changing activation precisions for early stages can indeed bring substantial improvements in compute efficiency (so the extra efforts for kernels implementation can be justified), and providing some additional simple experimental results will be preferred.

**Questions:**

How is $Bit_{act}$ updated in Algorithm 1 with respect to t? It seems like the intervals are always multiples of 0.05T. Was $Bit_{act}$ only updated every 0.05T?

**Limitations:**

The authors adequately addressed the limitations.

---

> ### Author Rebuttal · Authors · 2023-08-10
>
> Thanks for the constructive comment.
>
> We will now discuss 1) the practicality of the proposed RAQ, and 2) the granularity of Bit_act update.
>
> **Response ZFyZ-1: The practicality of the proposed RAQ.**
>
> As the reviewer rightly pointed out, we agree that accelerating diffusion models quantized with irregular bit widths on a GPU poses a challenge in terms of performance improvement in real world scenarios. Nonetheless, we believe that the proposed RAQ provides a meaningful direction for advancing the acceleration of diffusion models. Let us discuss more in detail below.
>
> First, while we agree that accelerating diffusion models quantized with irregular bit widths (e.g., 6 bits) on a GPU is indeed challenging due to the lack of corresponding arithmetic units, we would like to emphasize that our main contribution is to show a direction that we can reduce bit resolution for a part of parameters, and our approach is not fundamentally limited to a certain bit resolution. Hence, if other works that can further reduce the overall bit resolution of the network are developed independent of our scheme, then it can be combined with our proposed scheme, and there is a chance that portion of the 4-bit parameters in our scheme can be increased to see realistic performance benefits. For example, our current approach involves a basic min/max-based quantization mechanism for activation quantization, leading to an 8-bit quantization for the baseline fixed bitwidth case. Then, the allocation of activation bits using the RAQ method is distributed as (4b, 5b, 6b, 7b, 8b) = (20%, 20%, 10%, 40%, 10%) for LSUN-churches (Table 1). However, in the scenario where an advanced quantization mechanism enables 6-bit activation quantization even for the baseline fixed bitwidth case, there exists the possibility of increasing the proportion of 4-bit activations with the proposed RAQ, such as (4b, 5b, 6b) = (50%, 30%, 20%).
>
> Second, while accelerating diffusion models quantized with irregular bit widths (e.g., 6 bits) on a GPU is indeed challenging, the utilization of specialized hardware, like bit-scalable accelerators, could offer a promising solution for processing these models [1]. These accelerators are purpose-built to harness the benefits of quantization on a bit-by-bit basis, resulting in a nearly linear improvement in computing efficiency, encompassing both latency and energy consumption, with decreasing bit width. We introduce the concept of Bit Operations (BOPs), where BOPs are calculated as the product of OPs with weight bitwidth and activation bitwidth. This metric allows us to estimate the performance gain achievable through the RAQ method. In the case of LSUN-Churches, the full-precision baseline and Q-diffusion necessitates 4285.2 and 148.6 TBOPs (TBOPs: Tera BOPs) for a single image generation respectively, while the proposed RAQ method only requires 108.8 TBOPs. This indicates that with the utilization of specialized accelerators, the implementation of the RAQ approach could potentially lead to more than 39.4 times speedup and energy savings compared to the full-precision baseline, while Q-diffusion can achieve 28.8 times improvement compared to the full-precision baseline.
>
> Table A-1. LSUN-Churches (256x256) generation results
> | Model |W/A | FID | TBOPs|
> |---|---|---|---|
> | LDM-8 |32/32 | 4.09 | 4285.2 |
> | Q-diffusion | 4/8 | 4.45 | 148.6 |
> | RAQ | 4/6 | 4.64 | 108.8 |
>
> Table A-2.LSUN-Bedrooms (256x256) generation results
> | Model |W/A | FID | TBOPs|
> |---|---|---|---|
> | LDM-4 |32/32 | 2.96 | 20725.8 |
> | Q-diffusion |4/8 | 4.17 | 681.2 |
> | RAQ |4/6 | 3.99 | 504.8 |
>
> [1] Fu, Yonggan, et al. "2-in-1 accelerator: Enabling random precision switch for winning both adversarial robustness and efficiency." MICRO-54: 54th Annual IEEE/ACM International Symposium on Microarchitecture. 2021.
>
> **Response ZFyZ-2: The granularity of Bit_act update.**
>
> In the context of the RAQ method outlined in Algorithm 1, choosing a finer granularity for Bit_act update necessitates a larger number of sampled images for the optimization process. Meanwhile, our investigation revealed that consecutive timesteps within a 0.05T range exhibit similar sensitivity to noise injection. Therefore, our experimental configuration concentrated on optimizing the activation bitwidth with intervals of 0.05T for the Bit_act update. This approach ensures effective optimization while managing computational demands.
>
> Thanks again for the comments.

---

> > ### Comment · Reviewer_ZFyZ · 2023-08-19
> > **Response to Authors**
> >
> > Dear Authors,
> >
> > Thank you for the comprehensive rebuttal and the clarifications provided. The discussion of bit-scalable accelerators and some potential implications of the RAQ method resolve some of my concerns. I still incline to think the application scenarios of RAQ in real life are relatively limited (echoing with Reviewer eumA), but after careful consideration, I think this work demonstrates an intricate property of the early-stage robustness in diffusion models and provides empirical validation of how to utilize this property to benefit quantization, which is indeed novel and could inspire future research in the NeurIPS community. Thus, I decided to raise my rating slightly.
> >
> > Best regards,
> > Reviewer ZFyZ

---

### Official Review · Reviewer_rGNp · 2023-07-12

**Soundness:** 4 excellent
**Presentation:** 4 excellent
**Contribution:** 4 excellent
**Rating:** 5
**Confidence:** 3

**Summary:**

The author initially notes that errors in the early stages of the reverse diffusion process result in minimal disturbance to the final generated image. As a solution, they suggest employing low-bit activations for the initial reverse diffusion process while preserving high-bit activations for the subsequent stages, in conjunction with PTQ.

**Strengths:**

- The idea is clear and easy to understand

- The proposed RAQ method outperforms the other methods such as Q-diffusion


**Weaknesses:**

- Could you the authors explain how is the entropy calculated and why higher randomness in the pixel values will cause the images blurrier?


- Cpmparison to other methods. The authors mentioned two PTQ methods PTQ4DM and Q-diffusion, but only provide quantitative and qualitative comparison to baseline and Q-diffusion.


- In section 3.2, it seems obvious that add the same amound of noise to a noisier image will have less influence than to a less noiser image?

- And the authors did not explain why in figure 3, the performance on two different dataset are so different.

**Questions:**

- Why is the images in Figure (a) is mirrored?

- What does the term W4A6/8 mean in Figure 7

**Limitations:**

See weaknesses.

---

> ### Author Rebuttal · Authors · 2023-08-10
>
> Thanks for the constructive comment.
>
> We will now discuss 1) relationship between entropy and image clarity, 2) exclusion of comparison with PTQ4DM, 3) the effects of noise addition on images with varied clarity levels, 4) the performance difference between two datasets in Figure 3, 5) correction of mirrored images in Figure 1(a), and 6) clarification  of notation “W4A6/8” in Figure 7
>
> **Response rGNp-1: Relationship between entropy and image clarity**
>
> Entropy quantifies the level of randomness or uncertainty within a dataset, and when applied to image generation, it reflects the diversity or randomness in the pixel values of the generated images.The entropy of the random variable $X$ is defined as following equation [1]:
>
> $H(X)=\sum_{x}-p(X)\log p(X)$
>
> Higher entropy indicates a wider range of pixel values, which can introduce more noise and randomness to the images. This noise can lead to fluctuations in pixel values that do not align with the actual image structure, ultimately causing a decrease in image sharpness and clarity. In contrast, lower entropy suggests more structured and consistent pixel values, contributing to clearer and sharper images.
>
> For detailed information regarding the calculation of entropy within the context of our study, please refer to Supplementary Material A.2.
>
> [1] Shannon, Claude Elwood. "A mathematical theory of communication." The Bell system technical journal 27.3 (1948): 379-423.
>
> **Response rGNp-2: Exclusion of comparison with PTQ4DM**
>
> There are two key reasons behind our decision not to include a comparison with PTQ4DM in this study.
>
> Firstly, PTQ4DM primarily concentrates on quantizing both weights and activations to 8 bits. In contrast, the Q-diffusion approach achieves a more advanced level of quantization by reducing the bitwidth of weights to 4 bits while retaining activations at 8 bits. This advancement positions Q-diffusion as a state-of-the-art quantization technique for diffusion models.
>
> Secondly, PTQ4DM focuses on low-resolution image generation using DDIM-based models. The reported FID results in their work correspond to image resolutions such as 32x32 for CIFAR-10 and 64x64 for ImageNet. On the other hand, our paper places its focus on generating high-resolution images, specifically at a resolution of 256x256 for LSUN images and 512x512 for Stable Diffusion. The significant discrepancy in image resolution creates challenges in directly comparing our method and PTQ4DM due to the absence of compatible data points.
>
> **Response rGNp-3:  The effects of noise addition on images with varied clarity levels**
>
> As the reviewer correctly indicated, the analysis presented in Section 3 leads us to an additional conclusion: the influence of adding noise to already noisy images is relatively less significant compared to adding noise to images that possess greater clarity. These conclusions stem from  two main insights from our analysis. Firstly, images generated during the initial phases of the diffusion process show heightened noise levels compared to those generated later. Secondly, the early-stage diffusion process exhibits a higher resilience against the introduction of noise.
>
> Hence, based on these insights, we can reasonably infer that a more aggressive quantization approach could be applied during the early stages of the process, as more aggressive quantization causes higher quantization noise. This strategic choice aligns with the inherent strengths of the early diffusion steps and their ability to accommodate higher quantization noise resulting from more aggressive quantization.
>
> **Response rGNp-4:  The performance difference between two datasets in Figure 3**
>
> We believe that the variation in performance between the two datasets is attributed to the significant difference in their image resolutions. Specifically, the CIFAR-10 dataset comprises images with a resolution of 32x32, while the LSUN-churches dataset contains images at a higher resolution of 256x256. The increased resolution of LSUN-churches images introduces finer details into the generation process, making the image generation process more susceptible to noise. However, it's noteworthy that both datasets exhibit a similar early-stage resilience to noise injection, as demonstrated in our analyses.
>
> **Response rGNp-5:  Correction of mirrored images in Figure 1(a)**
>
> We appreciate your observation regarding the mirrored images in Figure 1(a). It appears that the mirroring occurred unexpectedly during the figure preparation process.
> We will correct the mistake and update the images in the final version of the paper. Thank you for bringing this to our attention.
>
> **Response rGNp-6:  Clarification  of notation “W4A6/8” in Figure 7**
>
> We acknowledge the concern you raised regarding the notation "W4A6/8" in Figure 7. We aimed to convey that the activation bits of the stable diffusion model were combined with both 6 bits and 8 bits precision. However, we recognize that this notation might lead to confusion among readers.
> To address this issue, we will update the explanation of the "W4A6/8" notation in the caption of the Figure 7 as follows: “Here, W4 denotes adopting 4-bit weights, A$n$ denotes $n$-bit activations, and A$n/m$ indicates adopting $n$-bit activations in the early stage of the diffusion stages, while the later stage adopts $m$-bit activations”. We appreciate your valuable feedback, and thank you for bringing this to our attention.
>
> Thanks again for the comments.

---

### Official Review · Reviewer_eumA · 2023-07-25

**Soundness:** 3 good
**Presentation:** 3 good
**Contribution:** 2 fair
**Rating:** 4
**Confidence:** 4

**Summary:**

In this submission, the authors propose a novel approach to speed up the noise estimation network by leveraging the robustness of early-stage diffusion models. Specifically, they present an algorithm to modify the quantization bit width according to the diffusion step. The proposed method shows positive results in reducing activation bits below 8 bits.

**Strengths:**

•	The writing of this manuscript is easy to follow, and the illustrations are clear.

•	This work is well-motivated. Based on the analysis, the authors provide insights on the different roles that different diffusion steps play and show the room to improve the PTQ process by treating early and later steps differently.

•	The experiments show positive results of the proposed method.


**Weaknesses:**

•	The real-world benefits of reducing activation bits. With advanced samplers, the sampling steps of diffusion models are significantly reduced, e.g., to 50 steps or lower. Thus, the gain achieved through low bit width calculation in the early steps may be marginal in real-world evaluation. On the other hand, bit width is usually a power of two. To my knowledge, some execution cores are designed to process 8-bit-only or 4-bit-only data. Irregular bit widths like 6 bits are treated as standard bit widths by padding zeros. Thus, the benefits of reducing to irregular bit widths (e.g., 6 bits) instead of standard bit widths (e.g., 4 bits) are questionable from the perspective of hardware. The authors are encouraged to provide real-world evidence of the benefits of RTQ or a discussion of the above concerns.

•	The choice of FID threshold. In the RTQ algorithm, the choice of FID threshold is critical since it determines the final bit width dictionary and thus the quantization gain. How do you set this hyperparameter for a new dataset?


**Questions:**

•	How does RTQ perform in accelerating diffusion models in real-world scenarios?

•	How do you set a reliable FID threshold for the RTQ algorithm?


**Limitations:**

The choice of FID threshold in the RTQ algorithm is unclear.

---

> ### Author Rebuttal · Authors · 2023-08-10
>
> **Response eumA-1: The influence of RAQ on accelerating diffusion models**
>
> As the reviewer rightly pointed out, advanced samplers have been recently presented to reduce the sampling steps, and we agree that accelerating diffusion models quantized with irregular bit widths on a GPU poses a challenge in terms of performance improvement in real world scenarios. Nonetheless, we believe that the proposed RAQ provides a meaningful direction for advancing the acceleration of diffusion models. Let us discuss more in detail below.
>
> First, the approaches of reducing sampling steps and the proposed RAQ are orthogonal methodologies for accelerating diffusion models unless an extreme reduction in sampling steps is undertaken, such as employing a single-step sampling strategy. Recent works for reducing sampling steps typically fine-tune diffusion models so that multiple sequential sampling actions can be encompassed by fewer sampling steps. Even in such scenarios, the fundamental property of diffusion models, where Gaussian noise progressively transforms into desired images across sampling steps, remains applicable. This intrinsic characteristic enables each step to display unique responses to quantization, thereby making our concept applicable even in situations with diminished sampling steps.
>
> Second, while we agree that accelerating diffusion models quantized with irregular bit widths (e.g., 6 bits) on a GPU is indeed challenging due to the lack of corresponding arithmetic units, we would like to emphasize that our main contribution is to show a direction that we can reduce bit resolution for a part of parameters, and our approach is not fundamentally limited to a certain bit resolution. Hence, if other works that can further reduce the overall bit resolution of the network are developed independent of our scheme, then it can be combined with our proposed scheme, and there is a chance that portion of the 4-bit parameters in our scheme can be increased to see realistic performance benefits. For example, our current approach involves a basic min/max-based quantization mechanism for activation quantization, leading to an 8-bit quantization for the baseline fixed bitwidth case. Then, the allocation of activation bits using the RAQ method is distributed as (4b, 5b, 6b, 7b, 8b) = (20%, 20%, 10%, 40%, 10%) for LSUN-churches (Table 1). However, in the scenario where an advanced quantization mechanism enables 6-bit activation quantization even for the baseline fixed bitwidth case, there exists the possibility of increasing the proportion of 4-bit activations with the proposed RAQ, such as (4b, 5b, 6b) = (50%, 30%, 20%).
>
> Third, while accelerating diffusion models quantized with irregular bit widths (e.g., 6 bits) on a GPU is indeed challenging, the utilization of specialized hardware, like bit-scalable accelerators, could offer a promising solution for processing these models [1]. These accelerators are purpose-built to harness the benefits of quantization on a bit-by-bit basis, resulting in a nearly linear improvement in computing efficiency, encompassing both latency and energy consumption, with decreasing bit width. We introduce the concept of Bit Operations (BOPs), where BOPs are calculated as the product of OPs with weight bitwidth and activation bitwidth. This metric allows us to estimate the performance gain achievable through the RAQ method. In the case of LSUN-Churches, the full-precision baseline and Q-diffusion necessitates 4285.2 and 148.6 TBOPs (TBOPs: Tera BOPs) for a single image generation respectively, while the proposed RAQ method only requires 108.8 TBOPs. This indicates that with the utilization of specialized accelerators, the implementation of the RAQ approach could potentially lead to more than 39.4 times speedup and energy savings compared to the full-precision baseline, while Q-diffusion can achieve 28.8 times improvement compared to the full-precision baseline.
>
> Table A-1. LSUN-Churches (256x256) generation results
> | Model |W/A | FID | TBOPs|
> |---|---|---|---|
> | LDM-8 |32/32 | 4.09 | 4285.2 |
> | Q-diffusion | 4/8 | 4.45 | 148.6 |
> | RAQ | 4/6 | 4.64 | 108.8 |
>
> Table A-2.LSUN-Bedrooms (256x256) generation results
> | Model |W/A | FID | TBOPs|
> |---|---|---|---|
> | LDM-4 |32/32 | 2.96 | 20725.8 |
> | Q-diffusion |4/8 | 4.17 | 681.2 |
> | RAQ |4/6 | 3.99 | 504.8 |
>
> [1] Fu, Yonggan, et al. "2-in-1 accelerator: Enabling random precision switch for winning both adversarial robustness and efficiency." MICRO 2021.
>
> **Response eumA-2: FID threshold setting for the RAQ algorithm**
>
> The RAQ algorithm employs the FID of diffusion models without activation quantization as the FID threshold. This aligns with the central objective of the RAQ algorithm, which is to adaptively vary the activation quantization bitwidth across the different sampling steps while preserving the quality of image sampling.
>
>  For example, when applying the RAQ algorithm to the LSUN-Churches dataset, we initially compute the FID of the LSUN-Churches dataset without activation quantization while generating 5,000 samples. This approach ensures that the optimization of activation quantization maintains FID at an acceptable level. Consequently, as illustrated in Table 1, the proposed RAQ technique achieves an effective activation bitwidth of 6 without compromising FID.
>
> On the other hand, we can achieve lower activation bitwidth by slightly compromising FID. In this case, it becomes necessary to make adjustments to the FID threshold. The amount of the adjustment is dependent on the predetermined level of FID tolerance. For example, if a 10% increase in FID is acceptable, then the FID threshold is increased by 10%. For example, in the context of LSUN-church cases, we achieved an effective activation bit count of 5.60, resulting in an FID of 5.12. This FID value represents an approximate 10% increase compared to that of the full precision activation bit configuration, where the FID is 4.45 (Table 1).
>
> Thanks for the comments.

---

> > ### Comment · Reviewer_eumA · 2023-08-17
> >
> > After reading author's rebuttal and other reviews, I have decided to keep my rating. I am satisfied with author's explanation on the choice of FID, which should be clarified in future versions. However, the practical concerns regarding sampling steps and irregular bit width remain as crucial weaknesses of this submission. I know RAQ is applicable to advanced samplers. My point is the benefit in running time is marginal when there are very few steps. On the other hand, theoretical measurements like FLOPs, MACs or BOP that the authors introduced, make sense when the underlying execution kernels are consistent. Most existing execution kernels are not precision-scalable and quantization algorithm should be evaluated in that scenario. In summary, I believe the authors propose a simple and effective algorithm in theory, but with limited evaluations in terms of practical value.

---

### Decision · Program_Chairs · 2023-09-21

**Decision:**

Accept (poster)

**Comment:**

This paper presents a robustness-aware quantization (RAQ) method to speed up diffusion models by finding that the early-stage denoising steps can be done with fewer bit in activation. The paper received five reviews, most of which are neutral positive (except that one review is 4).

In the paper, the finding that the inaccurate activation (or activation with fewer bits) in early stages of diffusion steps does not affect the accuracy of diffusion model is novel. Based on this finding, the paper proposed a robustness-aware quantization algorithms which dynamically adjusts the bits for activations during the diffusion steps. The rebuttal clarified most of the questions raised by the reviewers. The main remaining concern from the reviewers is the practicality of the proposed RAQ methods on GPUs (since varying bit widths is not easy to implement on current GPU architectures). While this is a valid point especially for practical usage, as the authors pointed out, the paper showed a direction to reduce bit resolution for future new GPU  architectures. Moreover, the authors did provide some explanations and potential implementation with some specialized hardware (e.g., bit-scalable acelrators [1]).

 Considering all the reviews, the rebuttal, and the discussions, the AC decided to accept the paper.